# Association of Two Indices of Insulin Resistance Marker with Abnormal Liver Function Tests: A Cross-Sectional Population Study in Taiwanese Adults

**DOI:** 10.3390/medicina58010004

**Published:** 2021-12-21

**Authors:** Adi-Lukas Kurniawan, Chien-Yeh Hsu, Jane C.-J. Chao, Rathi Paramastri, Hsiu-An Lee, Amadou-Wurry Jallow

**Affiliations:** 1Research Center for Healthcare Industry Innovation, National Taipei University of Nursing and Health Sciences, 365 Ming-te Road, Beitou District, Taipei 112, Taiwan; 2Department of Information Management, National Taipei University of Nursing and Health Sciences, 365 Ming-te Road, Beitou District, Taipei 112, Taiwan; cyhsu@ntunhs.edu.tw; 3Master Program in Global Health and Development, College of Public Health, Taipei Medical University, 250 Wu-Hsing Street, Xinyi District, Taipei 110, Taiwan; 4School of Nutrition and Health Sciences, College of Nutrition, Taipei Medical University, 250 Wu-Hsing Street, Xinyi District, Taipei 110, Taiwan; rara.paramastri@gmail.com; 5Nutrition Research Center, Taipei Medical University Hospital, 252 Wu-Hsing Street, Xinyi District, Taipei 110, Taiwan; 6Department of Computer Science and Information Engineering, Tamkang University, 151 Yingzhuan Road, Tamsui District, New Taipei 251, Taiwan; billy72325@gmail.com; 7National Health Research Institutes, 35 Keyan Road, Zhunan Town, Miaoli County 350, Taiwan; 8Department of Health Technology, National Taipei University of Nursing and Health Sciences, 365 Ming-te Road, Beitou District, Taipei 112, Taiwan; amswurryjallow@gmail.com

**Keywords:** insulin resistance, liver function, triglyceride-glucose index, triglycerides to high-density lipoprotein cholesterol ratio

## Abstract

*Background and objectives:* Insulin resistance (IR) is frequently associated with chronic low-grade inflammation and has an important role as a mediator in the development of liver disease. Thus, this study aimed to explore the relationship between two indexes of IR and abnormal liver function parameters. *Materials and Methods:* This cross-sectional study obtained data of 41,510 men and 92,357 women aged ≥30 years from a private health screening institute in Taiwan. Two IR indexes namely triglyceride-glucose (TyG) index and triglycerides to high-density lipoprotein cholesterol (TG/HDL-C) ratio were used to examine their relationship to predict abnormal liver function parameters (aspartate aminotransferase (AST), alanine aminotransferase (ALT), gamma-glutamyl transferase (GGT), and alkaline phosphatase (ALP)). *Results:* Positive trend was shown for the association of TyG index in the highest quintile (Q5) and risk of high AST (OR = 1.45, 95% CI: 1.33–1.57), high ALT (OR = 1.85, 95% CI: 1.73–1.97), high GGT (OR = 2.04, 95% CI: 1.93–2.15), and high ALP (OR = 1.13, 95% CI: 1.07–1.19) compared with the median quintile (Q3) in the fully adjusted model. Similarly, participants in the Q5 of the TG/HDL-C ratio were associated with 1.38 (95% CI: 1.27–1.49), 1.71 (95% CI: 1.61–1.82), 1.75 (95% CI: 1.66–1.84), and 1.21 (1.16–1.27) odds for having high AST, ALT, GGT, and ALP respectively. The AUC (95% CI) value of the TyG index for predicting high AST, high ALT, and high GGT was 0.699 (0.692–0.705), 0.738 (0.734–0.742), and 0.752 (0.749–0.755), respectively. Meanwhile, the AUC (95% CI) of the TG/HDL-C ratio for predicting high AST, high ALT, and high GGT was 0.680 (0.673–0.686), 0.738 (0.734–0.742), 0.734 (0.731–0.738), respectively. *Conclusions:* Our study supported that the TyG index and TG/HDL-C ratio may be useful as non-invasive methods to predict the existence of impaired liver function in the early stage.

## 1. Introduction

Abnormal liver function determined by high plasma concentrations of liver enzymes is associated with type 2 diabetes (T2D) [1,2]. Insulin resistance (IR) remains the most robust indicator or progenitor of T2D and plays a critical role in the pathogenesis of diabetes-related comorbidities including cardiovascular disease and non-alcoholic fatty liver disease (NAFLD) [3]. The liver plays a key role in glucose metabolism by regulating various metabolic pathways such as glycolysis, glycogenesis, gluconeogenesis, and glycogenolysis [4]. Despite the leading-edge of liver disease diagnosis that has been primarily through biopsy or liver imaging, a non-invasive approach has been widely performed to assess liver abnormalities [5]. Alanine aminotransferase (ALT), aspartate aminotransferase (AST), gamma-glutamyl transferase (GGT), and alkaline phosphatase (ALP) are common major liver biomarkers used in clinical practice to assess impaired liver function [6]. A previous cross-sectional study has documented that NAFLD and liver biomarkers are significantly associated with three confirmed indices of hepatic IR [7]. Elevated IR is usually related to long-standing liver impairment and is a pathophysiological feature of hepatogenous insulin intolerance [8]. Researchers assumed that abnormal liver function augments hepatic IR to increase type 2 diabetes risk [9]. On the other hand, type 2 diabetes and IR lead to chronic hepatomegaly and immunological changes that enhance the abnormality of liver function [10]. However, it is not well known whether abnormal liver function leads to, originates from, or is simply related to IR and type 2 diabetes, as the underlying factors have not yet been elucidated [11,12].

A hyperinsulinemic-euglycemic clamp was recommended as the goal standard for quantifying IR. Since the clamp technique is costly and tedious, it is no longer durable for daily clinical practice [13]. Likewise, the homeostasis model assessment of insulin resistance (HOMA-IR) method, which calculates IR by integrating fasting glucose and insulin levels, is less invasive and tedious [14,15]. Although, the HOMA-IR indices are more widely used and beneficial in epidemiological and clinical studies for forecasting diabetes incidence in different groups. However, their application in clinical settings is minimal due to a lack of reference values for normal and abnormal insulin sensitivity [16].

Recently, the triglyceride–glucose (TyG) index and triglycerides to high-density lipoprotein cholesterol (TG/HDL-C) ratio have emerged as feasible tools for assessing IR in clinical therapy. The TyG index, defined as the product of fasting triglycerides and glucose levels, has been proposed as an ideal surrogate marker to evaluate IR [17,18]. The TyG index might represent an effective and simple tool to detect abnormal liver function when other procedures are unfitted [19]. Alternatively, the significance of the TG/HDL-C ratio in clinical practice has been investigated in many overwhelming studies. The procedure was considered simple and strongly recommended for the evaluation of insulin resistance [18,20], atherogenic dyslipidemia [20], and metabolic syndrome [21]. Other studies have also compared the TyG index and TG/HDL-C ratio with IR sensitivity. It has been reported that the TyG index and TG/HDL-C ratio both have a strong correlation with IR [22,23]. Despite the fact that insulin sensitivity might be further associated with impaired liver function, we found few studies demonstrating a relationship between the two indices of IR and abnormal liver function. Early detection of abnormal liver function could be essential for public health interventions and, more importantly, a simple and ideal diagnostic tool that allows timely detection and treatment. The TyG index and TG/HDL-C ratio might be feasible tools to assess abnormal liver function in large population-based epidemiological studies, as both techniques are easy, non-invasive, and less tedious compared to the old diagnostic techniques. The performance of the newly developed techniques such as the TyG index and TG/HDL-C ratio may also surpass the performance of the old techniques in terms of their poor reliability and reproducibility. Therefore, it would be necessary to develop and evaluate the use of the two indices of insulin resistance for the assessment of abnormal liver function in clinical and epidemiological studies. Since the two indices are strongly associated with IR and IR precedes metabolic syndrome as well as abnormal liver function. Thus, the present study was designed to investigate the association between the two indices of IR and abnormal liver condition.

## 2. Materials and Methods

### 2.1. Subjects and Study Design

The study participants were individuals who visited the Mei Jau Health Management (MJHM) Institution in Taiwan for an annual health examination from 2001 to 2015. All participants attended a standardized medical screening program in four MJHM health clinics throughout Taiwan (Taipei, Taoyuan, Taichung, and Kaohsiung), completed a validated self-administered questionnaire, underwent a physical examination, and provided blood samples for laboratory tests. In the present study, we initially selected women and men who were 30 years or older (*n* = 307,858; women *n* = 233,776, men *n* = 74,082). The exclusion criteria were: (1) had all types of cancer (*n* = 6080); (2) had hepatitis, cirrhosis, stroke, hyperthyroidism, and kidney disease (*n* = 41,464); and (3) had multiple entries (participants who had more than one annual health examination) (*n* = 126,447). A total of 41,510 men and 92,357 women (*n* = 133,867) met the inclusion criteria and were used for analysis. All participants provided informed consent for allowing their data to be processed, their identification was removed, and they remained anonymous throughout the study. The study protocols were approved by the Taipei Medical University Joint Institutional Review Board (TMU-JIRB N202010035).

### 2.2. Clinical Measurements

Anthropometry was obtained including weight and height (Nakamura KN-5000A, Tokyo, Japan), waist and hip circumference, percentage of body fat (Tanita TBF-410, Champaign, IL, USA), and blood pressure (BP) (Omron HEM-7201, Kyoto, Japan). Body mass index (BMI) was calculated as weight (kg)/height square (m) and classified as normal (18.5 kg/m^2^ ≤ BMI < 24 kg/m^2^) or overweight to obese (BMI ≥ 24 kg/m^2^) [24]. BP was measured twice at 10 min intervals after participants had been sitting for 5 min. Hypertension was defined as systolic/diastolic BP ≥ 140/90 or taking antihypertensive drugs [25]. Blood samples were collected after at least 8 h of overnight fasting and analyzed for the liver function test (aspartate aminotransferase (AST), alanine aminotransferase (ALT), gamma-glutamyl transferase (GGT), and alkaline phosphatase (ALP)), fasting blood glucose (FBG), triglycerides (TG), total cholesterol (TC), low-density lipoprotein cholesterol (LDL-C), high-density lipoprotein cholesterol (HDL-C), uric acid, creatinine, estimated glomerular filtration rate (eGFR), C-reactive protein (CRP), neutrophils, and lymphocytes. All blood samples were determined using an automated analyzer (Toshiba C8000, Tokyo, Japan) and performed at MJHM central laboratory with a coefficient of variation of less than 3%. The TyG index was calculated as described previously: TyG = Ln [TG (mg/dL) × FBG (mg/dL)/2] [17]. In addition, the TG/HDL ratio was calculated as TG (mg/dL)/HDL-cholesterol (mg/dL).

Diabetes was determined as FBG ≥ 126 mg/dL or taking antihyperglycemic drugs [26]. Hyperuricemia was defined as uric acid > 420 μmol/L in men and > 360 μmol/L in women [27]. Reduced renal function was considered as eGFR < 60 mL/min/1.73 m^2^ calculated according to chronic kidney disease epidemiology (CKD-EPI) equation [28]. High inflammation was considered as CRP ≥ 28.6 nmol/L or neutrophil-to-lymphocyte ratio (NLR) ≥ 3.0 [29]. High AST, ALT, GGT, and ALP were defined as AST > 35 IU/L, ALT > 40 IU/L, GGT > 30 IU/L, and ALP > 120 IU/L respectively [30,31].

### 2.3. Other Covariates

Other variables collected from the self-reported questionnaire were age, marital status (single or married), educational level (below or above university level), annual income, physical activity or sports status, sleep time, sleep status, smoking status, alcohol consumption, and presence of cardiovascular disease. Physical activity status (intensity (light, moderate, heavy, intense) and duration (hours per week)) was categorized as ‘active’ if participants had at least moderate intensity with a minimum duration of ≥1 to 2 h per week and ‘inactive’ if otherwise. Sleep status was categorized as ‘insomnia’ and ‘sleep well’ according to the previous study [32]. Cardiovascular status was defined as a history of cardiovascular disease (CVD) or taking CVD drugs. Food consumption was assessed using a standardized and validated semi-quantitative food frequency questionnaire (FFQ). Participants were asked how often and in what portions they consumed 22 food items in the past month (i.e., portions per day or week, from lowest to highest frequency). Three dietary patterns were identified using principal component analysis, and food items were retained in the pattern if the absolute cut-off value of a factor loading was ≥0.30 (Appendix A). Dietary pattern scores were calculated for each pattern by summing up the frequency of food items weighted by their factor loadings [33].

### 2.4. Statistical Analysis

All statistical analyses were performed using STATA version 26 (StataCorp LP, College Station, TX, USA). Data were presented as number (percentage) for categorical variables and as mean ± standard deviation (SD) for continuous variables. Differences between quintiles of the TyG index and TG/HDL-C ratio were assessed using the chi-square test for categorical variables and a general linear model for continuous variables. Binary robust logistic regression analysis was performed to calculate the odds ratio (OR) and 95% confidence intervals (CI) for abnormal liver function in the different TyG and TG/HDL-C quintiles. The 3rd quintile (Q3—the median quintile of the population) was used as the reference category because the cut-off point for the TyG index in previous studies [19,34] was between the Q3 range. Two adjustment models were applied: Model 1 was adjusted for age, gender, BMI, body fat, waist-to-hip circumference (WHR), marital status, education level, physical activity, annual income, smoking, alcohol consumption, sleeping status (condition and time), hypertension, diabetes, and CVD status. Model 2 was adjusted for all variables in model 1 plus hyperuricemia, reduced kidney function, high inflammation, TC levels, and LDL-C levels. Multivariable-adjusted logistic regression for abnormal liver function tests associated with the highest quintiles of TyG index and TG/HDL-C ratio, compared with median quintile was further estimated in subgroups analysis by gender, age group, BMI, hyperuricemia status, and high inflammatory status. Finally, receiver operating characteristic (ROC) curve analysis was performed to obtain the area under the curve (AUC) and to test the predictive power of the TyG index and TG/HDL-C ratio for abnormal liver function. The optimal cut-off point for the TyG index and TG/HDL-C ratio was determined using the value that represents the best specificity and sensitivity and referring to previous studies [19,34]. The ROC curve was plotted using SPSS version 23 (IBM Corp., IL, USA). A two-sided *p*-value < 0.05 indicates statistical significance.

## 3. Results

### 3.1. Characteristics of the Study Population

Table 1 presents the characteristics of the study population according to quintiles of the TyG index and TG/HDL-C ratio. Among a total of 133,867 participants, the overall prevalence of high AST, ALT, GGT, and ALP levels was 5.8% (*n* = 7712), 11.9% (*n* = 15,891), 18.0% (*n* = 24,091), and 21.4% (*n* = 28,588), respectively. All the characteristics differed significantly (*p* < 0.05) across the quintiles of both the TyG index and the TG/HDL-C ratio, except for the Western-style dietary scores in the TyG index quartiles. As expected, the prevalence of abnormal liver function tests was significantly increased with increasing quintiles of the TyG index and TG/HDL-C ratio (Table 1).

### 3.2. Association between TyG Index and TG/HDL-C Ratio with Abnormal Liver Function

Table 2 shows the adjusted ORs for the association between quintiles of the TyG index and abnormal liver function tests. A fully adjusted model showed that participants in the highest quintiles (Q5) of TyG index was positively associated with increased risk of high AST (OR = 1.45, 95% CI: 1.33–1.57), high ALT (OR = 1.85, 95% CI: 1.73–1.97), high GGT (OR = 2.04, 95% CI: 1.93–2.15), and high ALP (OR = 1.13, 95% CI: 1.07–1.19) compared to the median quintile (Q3). There was a similar trend of a positive association in the quintile 4 (Q4) of the TyG index for predicting abnormal liver function tests, except for high AST (OR = 1.07, 95% CI: 0.98–1.16, *p* = 0.125). The AUC (95% CI) value of the TyG index for predicting high AST, high ALT, and high GGT was 0.699 (0.692–0.705), 0.738 (0.734–0.742), and 0.752 (0.749–0.755), respectively. Using the optimal cut-off point of 8.50, the sensitivity and specificity for predicting high AST, high ALT, and high GGT were 67.8% and 61.7%, 72.0% and 64.2%, and 71.6% and 66.8%, respectively (Figure 1).

Similar to the TyG index, a positive association towards a higher risk of abnormal liver function tests was observed as quintiles of the TG/HDL-C ratio. However, the TG/HDL-C ratio showed lower ORs associated with abnormal liver function tests of AST, ALT, and GGT than the TyG index. A fully adjusted model revealed that participants in the Q5 of the TG/HDL-C ratio were associated with 1.38 (95% CI: 1.27–1.49), 1.71 (95% CI: 1.61–1.82), 1.75 (95% CI: 1.66–1.84), and 1.21 (1.16–1.27) odds for having high AST, ALT, GGT, and ALP, respectively (Table 2). Using the ROC curve, the optimal cut-off point for the TG/HDL-C ratio was 1.78, yielding sensitivity and specificity of 66.9% and 60.1% for predicting high AST, 66.9% and 60.1% for predicting high ALT, and 70.7% and 64.8% for predicting high GGT (Figure 1). The AUC (95% CI) of TG/HDL-C ratio for predicting high AST, high ALT, and high GGT was 0.680 (0.673–0.686), 0.738 (0.734–0.742), 0.734 (0.731–0.738), respectively. In contrast, the AUC value of both indices for predicting high ALP was relatively small (0.588 for TyG index and 0.590 for TG/HDL-C ratio), thus it generated lower sensitivity and specificity (TyG sensitivity and specificity: 49.6% and 61.9%, respectively, and TG/HDL-C ratio sensitivity and specificity: 51.1% and 61.1%). Moreover, compared with individual fasting blood glucose, triglyceride, and HDL-C levels, the AUC value of both TyG index and TG/HDL-C ratio was higher (Appendix A). Linear prediction of TyG index and TG/HDL-C ratio with serum liver function also showed a positive association (Appendix A).

### 3.3. Subgroup Analysis According to TyG Index and TG/HDL-C Ratio

We further divided the population into various subgroups and compared the predictive power of the TyG index and TG/HDL-C ratio. Subgroup analysis showed that the association of the TyG index and TG/HDL-C ratio for each abnormal liver function parameter remained robust in the highest quintile (Q5) of each subgroup studied compared with the median quintile (Q3) (Figure 2 and Figure 3). However, the adjusted ORs for high ALP in hyperuricemic subjects were insignificant in Q5 of the TyG index (OR = 1.05, 95% CI: 0.96–1.14, *p* = 0.809). The results of subgroup analysis in other quintiles were shown in Appendix A.

## 4. Discussion

Our data indicate that relatively new parameters, including the TyG index and the TG/HDL-C ratio as a surrogate of insulin resistance, might play a role in detecting abnormal liver function. We also demonstrated that a higher TyG index and TG/HDL-C ratio were positively associated with a higher risk of abnormal liver function after adjusting for potential confounders. We also analyzed the sensitivity and specificity of these two markers. Thus, we observed a similar result for the TyG index and the TG/HDL-C ratio in terms of detecting abnormal liver function.

Liver function is known to be related to insulin resistance and metabolic syndromes associated with elevated insulin, triglyceride, and glucose index [35]. A previous prospective cohort study has shown the potential role of the TyG index in identifying non-alcoholic liver function disease (NAFLD) [34]. In the present study, a higher TyG index indicated a higher risk of liver abnormalities. Similar to a previous cross-sectional study conducted on 10,761 Chinese over 20 years of age, the number of NAFLD was significantly increased with the increasing levels of TyG (OR = 6.3, 95%CI: 5.3–7.5, *p* for trend < 0.0001) [19]. Furthermore, in a survey of 6445 Chinese adults, which covered urban and rural areas between 2011 and 2013, a significant association was found between more severe insulin resistance, as indicated by higher TyG index, and a higher risk of impaired liver function in a fully adjusted model (OR = 2.04, 95% CI: 1.93–2.17, *p* < 0.0001) [15]. In our study, a higher ALT was observed in a group with higher TyG levels, which is consistent with a prior study that found an increase in ALT by 1.22 (95% CI: 1.21–1.24) IU for a one-unit increment of TyG [15].

Our study also remarked that the TyG index with AUC values of 0.699 (AST), 0.738 (ALT), and 0.752 (GGT) could be an effective biomarker to identify impaired liver function. A high predictive value of the TyG index was also observed in a cross-sectional study of 4784 adult participants to identify NAFLD and liver fibrosis (AUC of 0.761) [36]. Similarly, a previous study also found a high predictive value of TyG (AUC = 0.782, 95% CI: 0.77–0.79, *p* < 0.0001) with a sensitivity of 72.2% and specificity of 70.5% [19]. This finding is expected since the TyG index, generated from TG and FBG, takes into account the two main metabolic variables affected in fatty liver and is closely coincides with insulin resistance, the main pathogenesis of NAFLD [19]. Importantly, in the SAM study, TyG was found to have a better correlation with altered hepatic insulin due to its strong association with hepatic fat distribution [37]. NAFLD is closely related to obesity and metabolic syndrome and is characterized by an abnormal accumulation of triglycerides in the liver, which contributes to hepatic insulin resistance. Hepatic insulin resistance leads to excessive production of FBG and VLDL, which contain abundant serum TG [38]. In particular, our findings revealed that participants in the highest quintile (Q5) of the TyG index had substantially higher BMI, body fat percentage, FBG, TC, TG, and LDL-C. Thus, according to these observations, it is reasonable to use the TyG index as an efficient diagnostic method to detect abnormal liver function [19].

Furthermore, our study demonstrated the potential role of the TG/HDL-C ratio in predicting impaired liver function. Interestingly, the TG/HDL-C ratio showed a similar trend to the TyG index, although the odds were relatively lower. In a previous cohort study conducted between May 1994 to December 2003 in 9039 Japanese adults, a strong association was observed between the TG/HDL-C ratio and an increased incidence of fatty liver disease (OR = 1.55, 95% CI: 1.35–1.77, *p* < 0.0001 for men and OR = 2.72, 95% CI: 1.88–3.95, *p* <0.0001 for women) [39]. Consistently, a previous study involving 18,061 Chinese adults who underwent a health checkup between May 2013 and June 2014 also indicated the strong association between the TG/HDL-C ratio and liver biomarkers, which OR and prevalence of abnormal liver function progressively increased across the quartile of the TG/HDL-C ratio [40].

We further examined the sensitivity and specificity of the TG/HDL-C ratio in detecting abnormal liver function by investigating the AUC. Our recent study demonstrated a relatively high AUC for predicting high AST, high ALT, and high GGT of 0.680 (0.673–0.686), 0.738 (0.734–0.742), and 0.734 (0.731–0.738), respectively. In accordance with the results of a prior retrospective cohort study among non-obese Chinese, TG/HDL-C ratio independently indicated abnormal liver function with AUC of 0.70 (0.68–0.72) in men and 0.72 (0.70–0.75) in women [41]. Compared with other lipid parameters and markers of liver injury, the AUC of the TG/HDL-C ratio in a study of adult NAFLD patients was 0.79 for men and 0.85 for women [40]. Although the mechanism underlying the association between TG/HDL-C and abnormal liver function has not been fully elucidated, insulin resistance is a possible mediator. TG/HDL-C has been found to be essential for insulin sensitivity and cardiovascular risk assessment [42,43]. The increased TG/HDL-C ratio also indicates the presence of a small dense LDL cholesterol subclass (sdLDL), which is atherogenic lipoprotein, and plays a crucial role in insulin resistance and several chronic metabolic disorders [44,45]. Insulin resistance, therefore, facilitates liver injury by stimulating adiposity and lipolysis of TG in adipose tissue and the liver [36,45]. In addition, adiponectin may provide another link between TG/HDL-C and impaired liver function [46]. Prior investigations have shown that adiponectin elevates serum HDL-C and conversely decreases serum TG. Moreover, reduced adiponectin levels could lead to a higher TG/HDL-C ratio [19]. Further studies are needed to clarify whether adiponectin contributes to the association between TG/HDL-C and abnormal liver functions.

The strength of our study is the large sample size, comprehensive analysis, and subgroup analysis to determine which group generates better results. However, some limitations should be noted. First, our study is cross-sectional, therefore, we cannot establish the causal effect inference. Second, we did not directly measure insulin concentration. Moreover, imaging data and liver biopsy to confirm the abnormality of liver function were not available in the present study. However, the purpose of our study was to investigate the association between insulin resistance indices, including TyG and TG/HDL-C ratio, and abnormal liver function tests. Third, the FFQ in our study did not include information on calorie and nutrient intake, which could affect the association between abnormal liver function and insulin resistance index.

## 5. Conclusions

In conclusion, our study supported the evidence that the TyG index and TG/HDL-C ratio were strongly associated with the progression of abnormal liver function. The TyG index and TG/HDL-C ratio may be useful as non-invasive methods to predict the existence of abnormal liver function in the early stage.

## Figures and Tables

**Figure 1 medicina-58-00004-f001:**
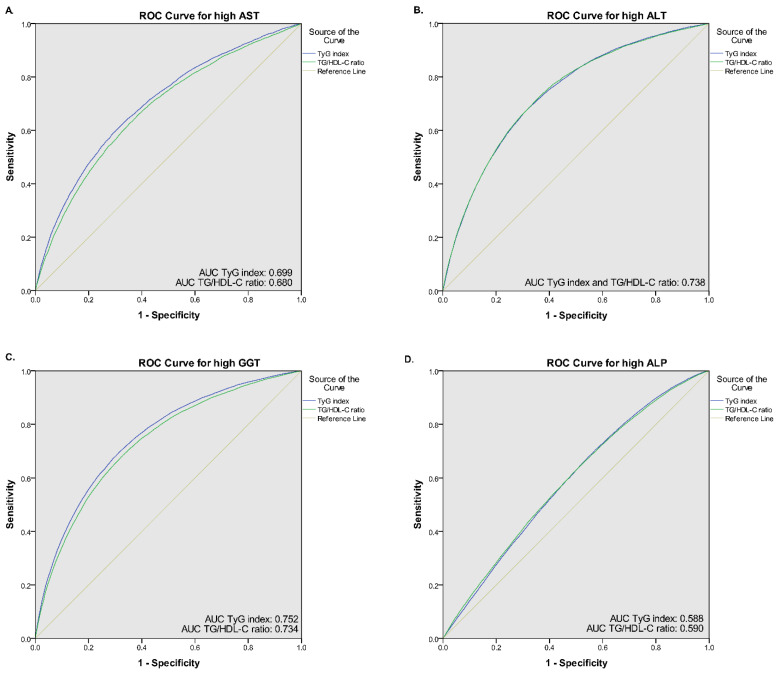
Receiver operative characteristic (ROC) curves of the TyG index and TG/HDL-C ratio for (**A**) high AST, (**B**) high ALT, (**C**) high GGT, and (**D**) high ALP.

**Figure 2 medicina-58-00004-f002:**
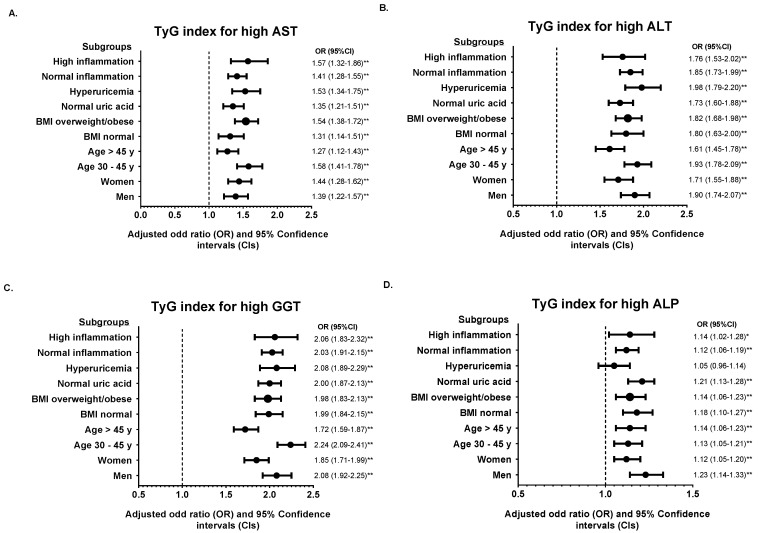
Subgroup analysis of the association between TyG index in quintile 5 (Q4) with (**A**) high AST, (**B**) high ALT, (**C**) high GGT, and (**D**) high ALP. Quintile 3 (Q3) was used for the reference. The odds ratio (OR) was adjusted with model 2 except for stratified variables in each subgroup. * *p* < 0.05, ** *p* < 0.01 for comparison of Q5 to Q3.

**Figure 3 medicina-58-00004-f003:**
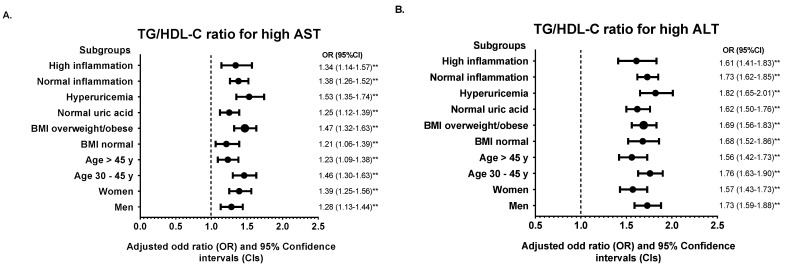
Subgroup analysis of the association between TG/HDL-C ratio in quintile 5 (Q4) with (**A**) high AST, (**B**) high ALT, (**C**) high GGT, and (**D**) high ALP. Quintile 3 (Q3) was used for the reference. The odds ratio (OR) was adjusted with model 2 except for stratified variables in each subgroup. ** *p* < 0.01 for comparison of Q5 to Q3.

**Table 1 medicina-58-00004-t001:** Characteristics of the participants according to quintiles of TyG index and TG/HDL-C ratio.

Variables	All	Quintiles of TyG Index					*p* ^a^	Quintiles of TG/HDL-C Ratio					*p* ^a^
		Q1 (4.73–7.89)	Q2 (7.90–8.20)	Q3 (8.21–8.51)	Q4 (8.52–8.90)	Q5 (8.91–11.86)		Q1 (0.04–0.87)	Q2 (0.88–1.27)	Q3 (1.28–1.84)	Q4 (1.85–2.91)	Q5 (2.92–19.95)	
*N*	133,867	26,762	26,786	267,40	26,806	26,773		26,779	26,775	26,768	26,783	26,762	
Gender							<0.001						<0.001
Men	41,510 (31.0)	3413 (8.2)	5996 (14.4)	8547 (20.6)	10,859 (26.2)	12,695 (30.6)		2706 (6.5)	5148 (12.4)	7954 (19.2)	11,120 (26.8)	14,582 (35.1)	
Women	92,357 (69.0)	23,349 (25.3)	20,790 (22.5)	18,193 (19.7)	15,947 (17.3)	14,078 (15.2)		24,073 (26.0)	21,627 (23.4)	18,814 (20.4)	15,663 (17.0)	12,180 (13.2)	
Age group							<0.001						<0.001
30–45 y	88,682 (66.2)	22,433 (25.3)	20,081 (22.6)	17,489 (19.7)	15,414 (17.4)	13,265 (15.0)		20,874 (23.5)	19,174 (21.6)	17,347 (19.6)	15,975 (18.0)	15,312 (17.3)	
>45 y	45,185 (33.8)	4329 (9.6)	6705 (14.8)	9251 (20.5)	11,392 (25.2)	13,508 (29.9)		5905 (13.1)	7601 (16.8)	9421 (20.9)	10,808 (23.9)	11,450 (25.3)	
Marital status ^b^							<0.001						<0.001
No	33,776 (26.2)	7822 (23.2)	7057 (20.9)	6527 (19.3)	6314 (18.7)	6056 (17.9)		7863 (23.3)	7129 (21.1)	6646 (19.7)	6210 (18.4)	5928 (17.5)	
Yes	95,054 (73.8)	17,908 (18.8)	18,686 (19.7)	19,248 (20.3)	19,492 (20.5)	19,720 (20.7)		17,859 (18.8)	18,638 (19.6)	19,107 (20.1)	19,585 (20.6)	19,865 (20.9)	
Educational attainment ^c^							<0.001						<0.001
Low	82,975 (62.5)	14,897 (18.0)	15,948 (19.2)	16,635 (20.0)	17,274 (20.8)	18,221 (22.0)		15,544 (18.7)	16,243 (19.6)	16,870 (20.3)	17,003 (20.5)	17,315 (20.9)	
High	49,821 (37.5)	11,680 (23.4)	10,621 (21.3)	9891 (19.9)	9320 (18.7)	8309 (16.7)		11,042 (22.2)	10,308 (20.7)	9681 (19.4)	9566 (19.2)	9224 (18.5)	
Annual income ^d^							0.023						<0.001
Low (<800,000 NTD)	74,266 (59.1)	14,990 (20.2)	15,057 (20.3)	14,858 (20.0)	14,671 (19.7)	14,690 (19.8)		15,230 (20.5)	15,232 (20.5)	15,024 (20.2)	14,640 (19.7)	14,140 (19.1)	
High (>810,000 NTD)	51,314 (40.9)	10,264 (20.0)	10,147 (19.8)	10,153 (19.8)	10,403 (20.3)	10,347 (20.1)		9937 (19.4)	9884 (19.3)	10,063 (19.6)	10,435 (20.3)	10,995 (21.4)	
Physical activity status							<0.001						<0.001
Inactive	74,021 (55.3)	15,897 (21.5)	15,101 (20.4)	14,441 (19.5)	14,197 (19.2)	14,385 (19.4)		15,898 (21.5)	15,174 (20.5)	14,625 (19.8)	14,227 (19.2)	14,097 (19.0)	
Active	59,846 (44.7)	10,865 (18.2)	11,685 (19.5)	12,299 (20.5)	12,609 (21.1)	12,388 (20.7)		10,881 (18.2)	11,601 (19.4)	12,143 (20.3)	12,556 (21.0)	12,665 (21.1)	
Sleeping time							<0.001						<0.001
< 6 h	30,418 (22.7)	5562 (18.3)	5722 (18.8)	6092 (20.0)	6339 (20.8)	6703 (22.1)		5864 (19.3)	5774 (19.0)	6122 (20.1)	6271 (20.6)	6387 (21.0)	
≥ 6 h	103,449 (77.3)	21,200 (20.5)	21,064 (20.3)	20,648 (20.0)	20,467 (19.8)	20,070 (19.4)		20,915 (20.2)	21,001 (20.3)	20,646 (20.0)	20,512 (19.8)	20,375 (19.7)	
Sleeping condition ^e^							<0.001						<0.001
Insomnia	82,648 (62.2)	17,272 (20.9)	16,883 (20.4)	16,416 (19.9)	16,213 (19.6)	15,864 (19.2)		17,465 (21.1)	17,174 (20.8)	16,677 (20.2)	15,989 (19.3)	15,343 (18.6)	
Sleep well	50,215 (37.8)	9306 (18.5)	9700 (19.3)	10,141 (20.2)	10,371 (20.7)	10,697 (21.3)		9106 (18.1)	9393 (18.7)	9877 (19.7)	10,616 (21.1)	112,23 (22.4)	
Smoker ^f^	27,727 (21.1)	3854 (13.9)	4582 (16.5)	5324 (19.2)	6268 (22.6)	7699 (27.8)	<0.001	3668 (13.2)	4212 (15.2)	5154 (18.6)	6282 (22.7)	8411 (30.3)	<0.001
Alcoholic drinker	16,105 (12.0)	2194 (13.6)	2624 (16.3)	3195 (19.8)	3597 (22.4)	4495 (27.9)	<0.001	2472 (15.4)	2598 (16.1)	3009 (18.7)	3610 (22.4)	4416 (27.4)	<0.001
Presence of diseases													
Hypertension	19,476 (14.6)	1043 (5.4)	2040 (10.5)	3385 (17.4)	5170 (26.5)	7838 (40.2)	<0.001	1539 (7.9)	2469 (12.7)	3687 (18.9)	5154 (26.5)	6627 (34.0)	<0.001
Diabetes	5824 (4.4)	121 (2.1)	213 (3.6)	379 (6.5)	884 (15.2)	4227 (72.6)	<0.001	276 (4.7)	455 (7.8)	872 (15.0)	1501 (25.8)	2720 (46.7)	<0.001
Cardiovascular	3667 (2.7)	386 (10.5)	519 (14.2)	671 (18.3)	836 (22.8)	1255 (34.2)	<0.001	487 (13.3)	558 (15.2)	742 (20.2)	856 (23.3)	1024 (28.0)	<0.001
Hyperuricemia	31,232 (23.3)	2116 (6.8)	3607 (11.6)	5353 (17.1)	8256 (26.4)	11,900 (38.1)	<0.001	2015 (6.4)	3558 (11.4)	5374 (17.2)	8215 (26.3)	12,070 (38.7)	<0.001
Reduced renal function	3642 (2.7)	182 (5.0)	376 (10.3)	597 (16.4)	955 (26.2)	1532 (42.1)	<0.001	279 (7.6)	458 (12.6)	674 (18.5)	913 (25.1)	1318 (36.2)	<0.001
High inflammation	22,120 (16.5)	3170 (14.3)	3589 (16.2)	4245 (19.2)	4941 (22.3)	6175 (28.0)	<0.001	2979 (13.5)	3643 (16.5)	4384 (19.8)	5192 (23.5)	5922 (26.7)	
Liver function status													
High AST	7650 (5.72)	561 (7.3)	763 (10.0)	1163 (15.2)	1692 (22.1)	3471 (45.4)	<0.001	641 (8.4)	814 (10.6)	1168 (15.3)	1795 (23.5)	3232 (42.2)	<0.001
High ALT	15,853 (11.8)	800 (5.0)	1342 (8.5)	2350 (14.8)	3948 (24.9)	7413 (46.8)	<0.001	863 (5.4)	1301 (8.2)	2276 (14.4)	4001 (25.2)	7412 (46.8)	
High GGT	24,035 (18.0)	1212 (5.0)	2152 (9.0)	3570 (14.9)	6037 (25.1)	11,064 (46.0)	<0.001	1478 (6.1)	2249 (9.4)	3692 (15.4)	6076 (25.3)	10,540 (43.8)	<0.001
High ALP	28,568 (21.3)	3393 (11.9)	5056 (17.7)	6080 (21.3)	6748 (23.6)	7291 (25.5)	<0.001	3564 (12.5)	5037 (17.6)	5837 (20.4)	6652 (23.3)	7478 (26.2)	<0.001
Dietary score													
Western style	9.8 ± 2.3	9.9 ± 2.2	9.8 ± 2.2	9.7 ± 2.3	9.8 ± 2.3	9.8 ± 2.4	0.385	9.8 ± 2.2	9.7 ± 2.2	9.7 ± 2.3	9.8 ± 2.3	10.0 ± 2.4	<0.001
Vege-seafood style	8.5 ± 1.9	8.5 + 2.0	8.4 ± 1.9	8.5 ± 1.9	8.5 ± 1.9	8.5 ± 2.0	<0.001	8.5 ± 1.9	8.5 ± 1.9	8.5 ± 1.9	8.5 ± 1.9	8.5 ± 1.9	0.002
American breakfast style	5.5 ± 1.4	5.5 ± 1.4	5.5 ± 1.4	5.5 ± 1.4	5.5 ± 1.4	5.4 ± 1.4	<0.001	5.5 ± 1.4	5.5 ± 1.4	5.5 ± 1.4	5.5 ± 1.4	5.4 ± 1.4	<0.001
Anthropometry													
BMI, kg/m^2^	23.0 ± 3.5	20.8 ± 2.5	21.7 ± 2.9	22.8 ± 3.2	24.1 ± 3.4	25.4 ± 3.5	<0.001	20.7 ± 2.5	21.7 ± 2.9	22.9 ± 3.2	24.1 ± 3.4	25.4 ± 3.4	<0.001
Body fat, %	27.8 ± 7.1	25.1 ± 5.5	26.3 ± 6.3	27.7 ± 7.0	29.2 ± 7.3	30.8 ± 7.5	<0.001	25.2 ± 5.6	26.7 ± 6.4	28.0 ± 7.1	29.2 ± 7.5	30.0 ± 7.4	<0.001
WHR	0.8 ± 0.1	0.7 ± 0.1	0.8 ± 0.1	0.8 ± 0.1	0.8 ± 0.1	0.9 ± 0.1	<0.001	0.7 ± 0.1	0.8 ± 0.1	0.8 ± 0.1	0.8 ± 0.1	0.9 ± 0.1	<0.001
Blood biochemistry													
FBG, mg/dL	99.4 ± 19.6	91.6 ± 6.9	94.6 ± 7.4	97.1 ± 8.5	100.1 ± 11.0	113.4 ± 36.4	<0.001	93.7 ± 9.9	95.8 ± 12.3	98.5 ± 16.3	101.7 ± 20.4	107.2 ± 29.4	<0.001
TG, mg/dL	105.8 ± 66.1	47.3 ± 8.4	67.5 ± 7.4	88.4 ± 10.1	120.7 ± 16.9	205.2 ± 77.6	<0.001	49.5 ± 11.1	68.7 ± 13.0	88.8 ± 17.1	119.3 ± 24.0	202.9 ± 79.0	<0.001
TC, mg/dL	194.8 ± 35.2	177.6 ± 30.1	186.2 ± 31.3	193.9 ± 32.1	202.9 ± 33.8	213.5 ± 36.7	<0.001	186.4 ± 31.9	188.0 ± 32.9	192.7 ± 34.2	200.0 ± 35.4	207.2 ± 36.8	<0.001
LDL-C, mg/dL	115.2 ± 31.7	99.5 ± 26.2	109.0 ± 28.3	117.1 ± 29.6	124.9 ± 31.4	125.5 ± 34.5	<0.001	100.7 ± 27.4	109.5 ± 28.8	117.2 ± 30.3	124.9 ± 31.6	123.7 ± 33.6	<0.001
HDL-C, mg/dL	59.1 ± 15.3	68.6 ± 14.9	63.9 ± 14.4	59.6 ± 14.0	54.8 ± 13.0	48.6 ± 11.3	<0.001	75.6 ± 13.9	65.0 ± 11.4	58.3 ± 10.4	52.1 ± 9.3	44.4 ± 8.3	<0.001
Insulin resistance indexes													
TyG index	8.4 ± 0.6	7.7 ± 0.2	8.1 ± 0.1	8.4 ± 0.1	8.7 ± 0.1	9.3 ± 0.3	<0.001	7.7 ± 0.2	8.1 ± 0.2	8.4 ± 0.2	8.7 ± 0.2	9.2 ± 0.4	<0.001
TG/HDL-C ratio	2.0 ± 1.7	0.7 ± 0.2	1.1 ± 0.3	1.6 ± 0.4	2.3 ± 0.7	4.5 ± 2.3	<0.001	0.7 ± 0.1	1.1 ± 0.1	1.5 ± 0.2	2.3 ± 0.3	4.7 ± 1.8	<0.001

^a^ *p* value was analyzed by chi-square test for categorical variables and general linier model for continuous variables. ^b^ Total *n* = 128,830. ^c^ Total *n* = 132,796. ^d^ Total *n* = 125,580. ^e^ Total *n* = 132,863. ^f^ Total *n* = 131,498. ALP, alkaline phosphatase; ALT, alanine aminotransferase; AST, aspartate aminotransferase; BMI, body mass index; FBG, fasting blood glucose; GGT, gamma-glutamyl transferase; HDL-C, high-density lipoprotein cholesterol; LDL-C, low-density lipoprotein cholesterol; TC, total cholesterol; TG, triglyceride; TyG, triglyceride-glucose index; WHR, waist to hip ratio.

**Table 2 medicina-58-00004-t002:** Multivariable adjusted logistic regression for abnormal liver function tests in different quintiles of TyG index and TG/HDL-C ratio.

	High AST		High ALT		High GGT		High ALP	
	Model 1	Model 2	Model 1	Model 2	Model 1	Model 2	Model 1	Model 2
TyG index								
Q1	0.83 (0.74–0.93) **	0.86 (0.77–0.97) *	0.65 (0.59–0.71) **	0.69 (0.63–0.75) **	0.56 (0.52–0.61) **	0.61 (0.56–0.65) **	0.55 (0.52–0.58) **	0.56 (0.53–0.59) **
Q2	0.85 (0.77–0.94) **	0.87 (0.78–0.96) **	0.76 (0.71–0.82) **	0.79 (0.73–0.85) **	0.76 (0.72–0.81) **	0.79 (0.75–0.84) **	0.83 (0.79–0.86) **	0.83 (0.79–0.87) **
Q3	1.00	1.00	1.00	1.00	1.00	1.00	1.00	1.00
Q4	1.13 (1.04–1.23) **	1.07 (0.98–1.16)	1.32 (1.24–1.41) **	1.25 (1.17–1.33) **	1.47 (1.40–1.54) **	1.35 (1.29–1.43) **	1.10 (1.05–1.15) **	1.07 (1.02–1.11) **
Q5	1.79 (1.66–1.94) **	1.45 (1.33–1.57) **	2.18 (2.05–2.31) **	1.85 (1.73–1.97) **	2.71 (2.58–2.85) **	2.04 (1.93–2.15) **	1.19 (1.14–1.25) **	1.13 (1.07–1.19) **
TG/HDL-C ratio								
Q1	0.98 (0.88–1.09)	0.91 (0.81–1.01)	0.82 (0.75–0.90) **	0.78 (0.71–0.85) **	0.72 (0.67–0.77) **	0.61 (0.57–0.66) **	0.59 (0.56–0.62) **	0.60 (0.57–0.64) **
Q2	0.91 (0.82–1.00)	0.90 (0.81–0.99) *	0.81 (0.75–0.87) **	0.80 (0.74–0.87) **	0.81 (0.77–0.87) **	0.79 (0.74–0.84) **	0.86 (0.82–0.90) **	0.87 (0.83–0.91) **
Q3	1.00	1.00	1.00	1.00	1.00	1.00	1.00	1.00
Q4	1.14 (1.05–1.23) **	1.10 (1.01–1.19) *	1.30 (1.22–1.38) **	1.25 (1.18–1.33) **	1.37 (1.30–1.44) **	1.32 (1.25–1.39) **	1.15 (1.10–1.20) **	1.11 (1.06–1.16) **
Q5	1.66 (1.53–1.79) **	1.38 (1.27–1.49) **	2.02 (1.90–2.14) **	1.71 (1.61–1.82) **	2.23 (2.12–2.34) **	1.75 (1.66–1.84) **	1.34 (1.28–1.41) **	1.21 (1.16–1.27) **

Data are expressed as beta (β) and 95% confidence intervals (CIs) in the parenthesis. Model 1: adjusted by age and gender, BMI, body fat, WHR, marital status, education level, physical activity status, income status, smoking, alcohol drinking, sleeping status (condition and time), hypertension, diabetes, and cardiovascular disease status. Model 2: adjusted by model 1 + hyperuricemia, reduced kidney function, high inflammation, T-Cholesterol, LDL-C levels, and all type of dietary pattern scores. * *p* < 0.05, ** *p* < 0.01 for comparison of Q1, Q2, Q4, and Q5 to Q3.

## Data Availability

The data that support the findings of this study are available from Mei Jau (M.J.) Health Management Institute, but restricted for research use only. The data are not publicly available. Data are available from the authors upon reasonable request and with permission of MJ Health Management Institute.

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
