# Peer review of "Association of Two Indices of Insulin Resistance Marker with Abnormal Liver Function Tests: A Cross-Sectional Population Study in Taiwanese Adults"

_medicina, 2021, doi:10.3390/medicina58010004_

Round 1

Reviewer 1 Report

minor revision of the english.

If the authors has also verify if there is a correlation with the HbA1c and/or protein c reactive with the TyG index and the TG/HDL-c ratio. 

Author Response

Dear reviewer,
We thank you for your comments and suggestions for improving the manuscript. The authors have considered the comments and revised the manuscript accordingly. Please see the attachment for the review report. 

Reviewer 2 Report

Summary

The study from Adi Lukas Kurniawan et al. describes a cross-sectional study based on large sample (n=133867; 41510 men and 92357 women) aged ≥30 years from a private health screening institute in Taiwan. They examined the relationship between abnormal liver function parameters and two IR indexes: triglyceride-glucose (TyG) index and triglycerides to high-density lipoprotein cholesterol (TG/HDL-C) ratio. The authors show Positive trend for the association of TyG index or TG/HDL-C ratio in the highest quintile (Q5) and risk of high abnormal liver function parameters. The scope of the study is clear, and the manuscript is well written. The statistical methods are straightforward and described in detail.

I have only one comment

  • I thank the authors for describing in detail how study population was selected from individuals who visited the Mei Jau Health Management (MJHM) Institution in Taiwan for an annual health examination from 2001 to 2015. After exclusion A total of 41510 men and 92357 women remain for analysis.

The men/woman ratio seems too much unbalanced. Is this ratio equivalent to the ratio before exclusion (n=307858)? If not, may be a selection bias was introduced and should be acknowledged as limitation.

Author Response

(The authors gave the same response as above.)
